# Developmental Changes of Duckling Liver and Isolation of Primary Hepatocytes

**DOI:** 10.3390/ani13111820

**Published:** 2023-05-31

**Authors:** Qiang Bao, Laidi Wang, Xiaodan Hu, Chunyou Yuan, Yang Zhang, Guobin Chang, Guohong Chen

**Affiliations:** 1Key Laboratory for Evaluation and Utilization of Poultry Genetic Resources, Ministry of Agriculture and Rural Affairs, Yangzhou University, Yangzhou 225009, China; dx120200144@stu.yzu.edu.cn (Q.B.); yzwanglaidi@163.com (L.W.); chitahuxd@163.com (X.H.); mz120201368@yzu.edu.cn (C.Y.); gbchang@yzu.edu.cn (G.C.); ghchen2019@yzu.edu.cn (G.C.); 2Joint International Research Laboratory of Agriculture and Agri-Product Safety, The Ministry of Education of China, Yangzhou University, Yangzhou 225009, China

**Keywords:** hepatocytes, duck, ALB, CD36, CK18

## Abstract

**Simple Summary:**

The liver is the main site of fat synthesis in poultry, which participates in metabolic regulation. Although ducks are important poultry with high economic value, studies showing the liver development and isolation of primary hepatocytes of ducklings is still limited. Based on this, we observed morphological changes in duckling livers from the embryonic period to the first week after hatching, and hepatocytes were isolated from 21-day-old duck embryos. These results showed that the weight of duckling livers increased with age from the embryonic period to the first week after hatching, and the hepatocytes that were cultured grew well. These were not only helpful to improving the understanding of liver development in vivo and hepatic cell cultures in vitro, but they also might further contribute to investigating lipid metabolism in ducks.

**Abstract:**

The liver is the main site of fat synthesis and plays an important role in the study of fat deposition in poultry. In this study, we investigated the developmental changes of duckling livers and isolated primary duck hepatocytes. Firstly, we observed morphological changes in duckling livers from the embryonic period to the first week after hatching. Liver weight increased with age. Hematoxylin-eosin and Oil Red O staining analyses showed that hepatic lipids increased gradually during the embryonic period and declined post-hatching. Liver samples were collected from 21-day-old duck embryos for hepatocyte isolation. The hepatocytes showed limited self-renewal and proliferative ability and were maintained in culture for up to 7 days. Typical parenchymal morphology, with a characteristic polygonal shape, appeared after two days of culture. Periodic acid-Schiff (PAS) staining analysis confirmed the characteristics of duck embryo hepatocytes. PCR analysis showed that these cells from duck embryos expressed the liver cell markers ALB and CD36. Immunohistochemical staining and immunofluorescence analysis also confirmed ALB and CK18 expression. Our findings provide a novel insight regarding in vitro cell culture and the characteristics of hepatocytes from avian species, which could enable further studies concerning specific research on duck lipid metabolism.

## 1. Introduction

The liver is a highly dynamic organ in the organism, with a diverse range of functions, including synthesis, metabolism, excretion, and detoxification processes [1,2]. It plays a major role in lipogenesis, providing lipids for all tissues and itself [3]. Studies have found that the majority of body fat in animals mainly comes from blood intake and synthesis in the liver [4]. In contrast to mammals, fat synthesis in birds occurs primarily in liver rather than adipose tissue [5,6,7]. In poultry, the fat synthesized by the liver accounts for more than 95% of the body fat [8], which may be related to the key role of the liver. In commercial poultry farming, the liver contributes more to fatty acids due to the lower fat content of the diet [9]. However, the massive deposition of fat can lead to oxidative stress, inflammation, and apoptosis of hepatocytes in the liver, which affect the growth and health in poultry [10]. According to previous studies, excessive fat deposition may be related to liver function damage during the feeding of meat ducks [11]. Under natural conditions, migratory birds experience non-pathological hepatic steatosis for energy storage before migration, which is reversed after lipid consumption [12,13]. However, ducks and geese, descendants of migratory birds, are often overfed to produce foie gras, providing a unique model of hepatic steatosis [14]. Thus, duck liver is a suitable model for studying lipid outward transport and fatty acid β-oxidation.

In the case of poultry, lipids necessary for organismal development are synthesized endogenously and predominantly unsaturated in the liver [15]. Then, the fatty acids synthesized are incorporated into triglycerides and transported from plasma to other tissues of the body in the form of lipoproteins. Previous studies also have shown that, after overfeeding, the rate of triglyceride (TG) synthesis in the liver is significantly higher than the rate of very low-density lipoprotein (VLDL) secretion [16]. Moreover, the transport channel of TG in the body becomes saturated, resulting in the rapid formation of fatty tissue in the liver due to an adequate supply of TG in the blood [17]. High-throughput sequencing results have revealed that acetyl-CoA carboxylase (ACC), fatty acid synthase (FASN), the elongation of very long-chain fatty acids protein-6 (Elovl6), stearoyl-CoA desaturase (SCD), and fatty acid desaturase 1 (FADS1) are involved in hepatocyte fatty acid synthesis [18]. Although significant progress has been made in understanding the mechanisms underlying the lipid metabolism in geese [18], while it was not clear in ducks.

The efficient isolation and culture of primary liver cells might play a pivotal role in further studies on both normal liver metabolism and liver disease. According to previous studies, most hepatocytes expressed Albumin (ALB) and Cytokeratin 18 (CK18) [19]. Among them, Albumin (ALB) is highly expressed in the liver and is responsible for a variety of biological functions due to its complex structure [20,21]. Cytokeratin 18 (CK18) is a cytoskeletal protein and a major intermediate filament family member expressed in the liver. Multiple studies have confirmed that CK18 is released in the liver as one of the most widely studied biomarkers in nonalcoholic steatohepatitis [22,23]. Moreover, fatty acid translocase, also known as CD36, is abundantly expressed in the liver [24,25,26]. Some studies have documented that CD36 is a well-recognized scavenger receptor for fatty acid uptake, which plays multiple physiological roles, including receptor-mediated free fatty acid uptake. Researchers have continuously improved the methods of isolating and culturing primary hepatocytes in human and animal models since 1967 [27,28,29,30]. However, access to primary hepatocytes can be limited, and therefore, other immortalized hepatocyte models are commonly used. Most of the liver cell lines used are cancerous cells that do not reflect the normal response to treatments.

The primary hepatocytes represent the gold standard for studying the mechanisms that control hepatic glucose, lipid, and cholesterol metabolisms in vitro. While most studies on duck hepatocytes follow the isolation method performed by Seglen [29], there are no special methods for their isolation. The traditional irrigation protocol is complicated to implement. Therefore, the objective of the current study was to determine the developmental changes in duck livers from embryonic periods to the first week after hatching and to provide novel insights for in vitro duck hepatocyte culture. This study aimed to provide a theoretical basis for the physiological characteristics of hepatic lipid metabolism in ducks.

## 2. Materials and Methods

### 2.1. Ethics Approval

All animal experiments were performed in accordance with the Regulations for the Administration of Experimental Animals issued by the Ministry of Science and Technology (Beijing, China). The experimental procedures involving animals were guided and approved by the Institutional Animal Care and Use Committee of Yangzhou University, with approval number 132-2020.

### 2.2. Animals and Liver Sampling

In this study, 58 Cherry Valley duck embryos were purchased from the Ecolovo Group in Jiangsu, China. Duck livers were isolated from embryos that were incubated for 14, 21, and 27 days, as well as from 7-day-old Cherry Valley ducks (*n* = 5). The livers were immediately measured for width and weight, with an accuracy of 0.01 g, using an electronic balance made by Sartorius, Germany. Two samples, each measuring 1 × 1 × 1 cm, were excised from the middle and lower parts of the right liver lobe of each bird. The middle part was then immersed in neutral buffered formalin (4%, pH 7, 20–24 °C) for subsequent histological analysis using H&E staining. Meanwhile, the lower part was quickly frozen in liquid nitrogen and stored at −80 °C for future experiments.

### 2.3. Liver Sections and Oil Red O Staining

Liver samples were dehydrated in an ascending graded series of ethanol, embedded in paraffin, serially sectioned at 5 μm, and stained with Meyer’s Hematoxylin and Eosin (H&E). The frozen liver samples underwent a series of procedures, including embedding, sectioning, and staining, all of which were performed by Wuhan Servicebio Technology Co., Ltd. in Wuhan, China. The stained sections were then observed under a light microscope for lipid analysis.

### 2.4. Isolation and Culture Procedures

Livers were excised from 21-day-old duck embryos and collected in preheated phosphate-buffered saline (PBS) (Hyclone, Logan, UT, USA). The liver was washed with PBS again and broken with tweezers to release the liver cells. The cells were then digested with 2 mg/mL collagenase type II (Gibco, Grand Island, NE, USA). The resulting mixture was filtered through 0.3 µm sieve screens and incubated at 37 °C for 20 min. To stop the digestion, cleaning fluids composed of RPMI 1640 medium (Gibco, USA), 10% fetal bovine serum (FBS) (Gibco, USA), and 1% penicillin-streptomycin (Solarbio, Beijing, China) were used. The mixture was filtered again through 0.300 µm and 0.0750 µm sieve screens and then centrifuged at 100× *g* for 5 min at room temperature (RT) to remove vascular-stromal cells. The resulting precipitate was resuspended using Red Blood Cell (RBC) Lysis Buffer (Solarbio, Beijing, China) for 10 min at room temperature. After centrifugation at 100× *g* for 5 min at RT, the precipitate was washed three times with cleaning fluids. The hepatocytes were then seeded at a density of 6 × 10^5^ cells/mL and cultured in growth medium (RPMI 1640 medium + 10% FBS + 10^−6^ mol/L dexamethasone + 10^−6^ mol/L insulin + 1% Penicillin-Streptomycin) at 37 °C with 5% CO_2_ in a humidified incubator. The morphological features of duck embryonic hepatocytes (DEHs) were observed using a microscope (Olympus, Tokyo, Japan).

### 2.5. Cell Proliferation Assays

Duck embryo hepatocytes were seeded in 24-well plates with a density of 2 × 10^5^ cells per dish. The cells were incubated for 0, 6, 24, 48, 72, 96, 120, and 144 h. After incubation, 50 µL of CCK-8 reagent (Vazyme, Nanjing, China) was added to each well, followed by incubation for 4 h at 37 °C. Absorbance at the 450 nm wavelength was measured to establish the growth curve of the cells, with a sample size of *n* = 6.

### 2.6. Periodic Acid Schiff (PAS) Staining

Duck embryo hepatocytes were seeded in 24-well plates, and after 48 h of incubation, the cells were fixed with 4% paraformaldehyde for 15 min. PAS staining was performed according to the manufacturer’s instructions (G1360, Solarbio, Beijing, China). Briefly, a periodic acid solution was added and oxidized at room temperature for 15–20 min. The plate was then washed twice with tap water and soaked twice in distilled water. The Schiff reagent was added and incubated in the dark at room temperature for 15 min. After that, the samples were rinsed with water for 2 min. Mayer’s hematoxylin-staining solution was added for 1 min. Images were acquired using a microscope (Olympus, Tokyo, Japan).

### 2.7. Reverse Transcription and PCR Verification

Total RNA was extracted from cultured cells using TRIzol reagent (Invitrogen, Carlsbad, USA). First-strand cDNA was synthesized from 1 μg of total RNA using a cDNA synthesis kit according to the manufacturer’s instructions (Takara Bio, Shiga, Japan), followed by 35 PCR cycles using a 2 × Taq Master Mix kit (Vazyme, Nanjing, China). β-actin was used as an internal control, and all assays were run in triplicate. Gene-specific primer pairs are listed in Table 1. The cycling conditions consisted of an initial denaturation step at 95 °C for 5 min, followed by 35 cycles at 95 °C for 15 s, 60 °C for 15 s, and 72 °C for 30 s. PCR products were visualized by electrophoresis on 2.0% (*w*/*v*) agarose gels.

### 2.8. Immunohistochemistry and Immunofluorescence

Immunohistochemical staining (IHC) was performed according to the manufacturer’s instructions (SABC-POD; SA1023, Boster, Wuhan, China). Briefly, a cell slide was fixed, and endogenous peroxidase was blocked with hydrogen peroxide. Anti-albumin monoclonal sheep serum (ARG23508, Arigo, Taiwan, China) at a dilution of 1:200 was used as the primary antibody. Biotinylated mouse anti-sheep IgG (RS2134, ImmunoWay, Plano, TX, USA) at a dilution of 1:200 was used as the secondary antibody, and mouse serum (10%) was added to prevent non-specific binding. The chromogen used was 3,3’-diaminobenzidine (AR1022, DAB, Boster, China) to detect the antigen–antibody complex. The cell slide was counterstained with Mayer’s hematoxylin and mounted on coverslips. Images were acquired using a microscope (Olympus, Tokyo, Japan).

Indirect immunofluorescence assays (IFA) were performed to identify DEHs. Cells were cultured for 48 h, fixed with 4% paraformaldehyde for 15 min, washed three times with PBS, permeabilized with 0.25% Triton X-100 for 15 min, and then washed twice in PBS. After blocking with 5% BSA-containing liquid (Boster, China) at room temperature for 30 min, the cells were incubated in 5% BSA-containing liquid (Boster, China) containing the rabbit polyclonal antibody anti-cytokeratin 18 (10830-1-AP, Proteintech, Chicago, IL, USA) at a dilution of 1:100 overnight at 4 °C. The following day, the antibody was removed, cells were thoroughly washed with PBS, and fluorescein isothiocyanate (FITC) Affinipure goat anti-rabbit IgG (1:100, Boster, China) was added to the samples and incubated at room temperature in the dark for 1 h. Nuclei were stained with 1 μg/mL DAPI in dH_2_O for 15 min and washed with PBS. Finally, fluorescence was observed using a fluorescence microscope (Olympus, Tokyo, Japan).

### 2.9. Statistical Analysis

The statistical analysis was conducted using GraphPad Prism 6 Software (GraphPad Prism 6, GraphPad Software, San Diego, CA, USA) and SPSS 22.0 (IBM SPSS: IBM SPSS Statistics Version 22.0). Results were expressed as mean ± SEM (standard error mean). The Gaussian distribution of the data was tested with the Kolmogorov-Smirnov (KS) test. To determine significant differences between the groups, a one-way ANOVA with a Tukey test was used, and *p* < 0.05 was considered statistically significant.

## 3. Results

### 3.1. Morphological Observations of Duck Livers

As shown in Table 2, the length and weight of duck embryo livers increased gradually with age (*p* < 0.001). The growth rate was higher during the first week post-hatch than during the embryonic period (*p* < 0.001). Liver width coincided with liver weight. Figure 1 for H&E and Oil Red O staining showed that the hepatic texture and reticular formations were not intact, and there was some interstice at embryonic day 14 (E14), suggesting that the liver was not fully developed. In addition, H&E staining results showed that small white cavities appeared at E21, increased until hatching day, and then decreased during the first week after hatching. Oil Red O staining also revealed that red circle drops increased from E14 to E27 and then declined at D7. These white cavities or red circle drops were fat droplets. Duck livers from embryonic day 21 with proper weight and fat droplets were selected to isolate the duck embryo hepatocytes.

### 3.2. Morphological Observation and Growth Curve of Cells

Primary cells isolated from duck embryo livers adhered to the plates within 6 h, started to aggregate and proliferate after one day, and tended to spread out, showing a typical polygonal cobblestone-like morphology after two days (Figure 2). After three days, the endothelial cells expanded rapidly and exhibited a fibroblast-like morphology in culture. Hepatocytes attached to them, and the mixed growth lasted for more than six days. As shown in Figure 3, the growth curves of the cells showed that the proliferation activity of the cells was the strongest on the second day and the fifth day. After the fifth day, the viability of the cells decreased significantly (*p* < 0.001). These results are consistent with the morphological observation results. We concluded that the hepatocytes were cultured for two days, grown well, exhibited typical parenchymal morphology, and had characteristic polygonal activity.

### 3.3. Detection of Glycogen in Duck Embryo Hepatocytes

The glycogen synthesis capacity of duck embryo hepatocytes cultured for 48 h was assessed using periodic acid-Schiff (PAS) staining. As shown in Figure 4, the duck embryo hepatocytes were positively stained with PAS, indicating the presence of glycogen and their normal function.

### 3.4. Expression Analysis of Function-Specific Genes of Duck Hepatocytes

Based on reverse transcription and PCR verification, the high expression of synthetic function-specific genes (ALB, CD36) was detected in duck embryo hepatocytes, while there was no expression in duck embryo fibroblasts (Figure 5A). Furthermore, immunohistochemical staining demonstrated that duck hepatocytes expressed ALB protein (Figure 5B). Immunofluorescence results showed that the hepatocytes were positive for CK18, whereas duck embryo fibroblasts were all negative (Figure 5C).

## 4. Discussion

As an important organ of the animal body, the liver is closely related to the endocrine and metabolic levels during growth and development [31]. Studies have found that the liver has multiple functions in the body, such as the secretion of bile and detoxification [1]. Currently, immortalized cell lines and primary cell isolates are important in vitro models widely used for hepatotoxicity tests [32]. In this study, we analyzed the developmental pattern of duckling livers during the embryonic period and the first week after hatching. Our findings indicated that the size and weight of the livers increased from E14 to the hatching day. We also observed rapid increases in hepatic lipid droplets from E14 to the hatching day using H&E and Oil Red O staining, which is consistent with the results in chicks [24]. During the first week after hatching, the weight of the liver increased rapidly, while the lipid droplets decreased gradually from D1 to D7. This phenomenon might be related to the enhancement of liver function. Among the studies found, ducklings take in more nutrients after hatching, especially protein and energy, which provide the necessary nutritional foundation for the growth of the liver. In addition, some studies also found that there are obvious changes in the endocrine and metabolic levels of the body during the early growth and development process [33]. These metabolic activities require the support and regulation of the liver, which further promotes the growth and development of the liver. This suggests that the liver plays a crucial role in providing energy for the rapid growth and development of ducklings in the immediate post-hatching period. Moreover, we found that the liver lobules are composed of hepatocytes (parenchymal cells) and non-parenchymal cells. Hepatocytes represent nearly 80% of the total liver volume and carry out several essential liver functions. These were consistent with observations in other mammals and poultry. Approximately 70–80% of the liver’s cells are hepatocytes, which play an important role in the vital activity of organisms [34]. Therefore, our results also provide a theoretical basis for understanding the physiological characteristics of liver development in ducklings from the embryonic period to the first week after hatching.

As is well known, hepatocytes are widely used in viral infection studies [27,30]. In past studies, duck hepatocytes were commonly used as a model for duck hepatitis B virus (DHBV) [35]. The isolation and culture of primary hepatocytes are very important for the establishment of in vitro liver models [36]. With the in-depth study of animal metabolism and lipid synthesis, hepatocytes are widely used. In the previous years, researchers attempted to isolate hepatocytes from the liver by mechanical and enzyme digestive methods [37]. However, these methods cause great damage to cells, and cells exhibited small numbers and low viability. A two-step procedure for collagenase perfusion was commonly used for hepatocyte isolation [29]. Previous reports have described the isolation of hepatocytes by portal vein perfusion from various species, including rats and mice [2,38,39]. In contrast to mammalians, the avian liver has much less connective tissue than the mammalian liver and lacks a true lobular structure [15,40]. Hepatocytes exhibit a lower differentiation ability and relatively simpler function during the embryonic stage compared to that during the early postnatal period [41,42]. Therefore, a simple hepatocyte separation method for avians is possible. Based on the yield and purity of hepatocytes, 21-day-old duck embryos were selected for hepatocyte isolation. The duck embryo hepatocytes were cultured for seven days, and we found that the hepatocytes had very limited proliferative capacity. After two days of culture, typical parenchymal morphology with a characteristic polygonal shape appeared, which was similar to the morphological structure of chicken hepatocytes [2] and macrophages from a mixed primary culture of bovine liver cells [43]. Furthermore, we found that cell viability reached its maximum with incubation on the second and fifth day, respectively. We speculate that duck embryo hepatocytes underwent a proliferation period of two days. After that, endothelial cells entered a logarithmic phase for five days, after which the cell’s activities began to decrease. In combination, morphological changes of duck hepatocytes were closely related to cell viability.

The liver is the main metabolic organ involved with fat, protein, and carbohydrates [3]. In chickens, primary hepatocytes are usually isolated from embryonic or adult chickens [44]. In addition, compared with broiler chickens, laying hens secrete a large amount of estrogen, which induces the liver to synthesize most of the lipids, and excess lipids seriously affect the activity of primary cells [45]. Therefore, we chose Cherry Valley meat ducks to isolate hepatocytes in this study. Beyond this, the ability of hepatocytes to store glycogen was confirmed through periodic acid-Schiff staining. As one of the main forms of energy storage, glycogen participates in the physiological functions and metabolic processes of hepatocytes [46]. Through the detection of glycogen, we could better understand the metabolic state and energy storage of cells. At the same time, the function of hepatocytes can be more effectively evaluated by detecting the content of glycogen in the process of separating and culturing hepatocytes [47]. Therefore, we observed that hepatocytes cultured for 48 h had good function. Hepatocytes were successfully isolated from duck embryos and verified using hepatocyte markers. In this study, fibroblast cells were used as a negative control. Fibroblast cells, as a common cell type, are easier to isolate and culture. At the same time, studies have demonstrated that fibroblast cells do not express CK18 [48], which allows the specific detection of hepatocytes by the fluorescent signal of CK18. Currently, there are few studies on the isolation of duck embryonic hepatocytes. According to the previous report of Jia et al., they selected 1-year-old Shaoxing ducks to investigate the replication of DHBV in primary duck hepatocytes [49]. In chickens, primarily tumorigenic cell lines, such as the immortalized chicken embryo liver cell line or embryonic liver cell cultures, have been utilized [50,51]. Primary duck embryonic hepatocytes are frequently employed for viral propagation [52]. Currently, only immortalized duck embryo liver mesenchymal cells are available from the ATCC, and there are no parenchymal duck liver cell lines. In conclusion, the hepatocytes we isolated during the embryonic period were in good condition and function normally. This study benefits the further development of related work on duck lipid metabolism and virus infection in the future.

## 5. Conclusions

In this study, we investigated the developmental changes of duckling livers and isolated primary duck hepatocytes. The morphological changes showed that the weight of duckling livers increased with age from the embryonic period to the first week after hatching. At the same time, hepatic lipids increased gradually during the embryonic period and declined post-hatching. Moreover, the hepatocytes that were cultured grew well. These results provide novel insights for the characteristics of hepatocytes and also lay a theoretical foundation for clarifying the regulatory mechanism of duck lipids.

## Figures and Tables

**Figure 1 animals-13-01820-f001:**
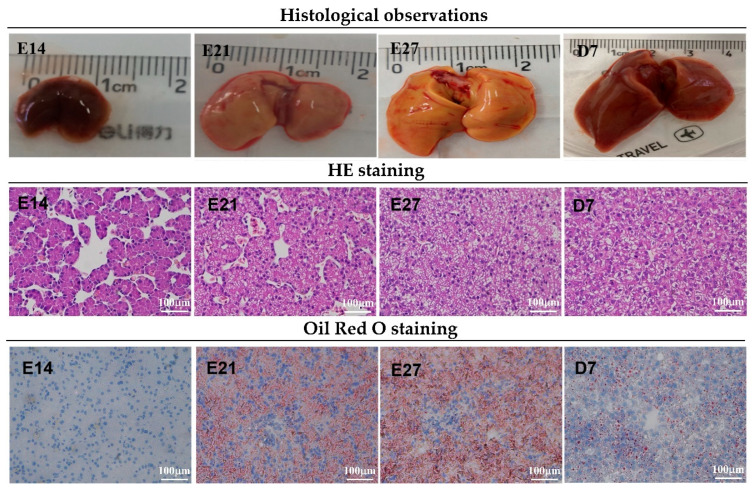
Liver morphology of ducklings from embryonic day 14 to post-hatch day 7. First row of pictures: liver size analysis. Second row: hepatic hematoxylin-eosin staining analysis. White cavities with different sizes are fat droplets, located in the cytoplasm. Third row: hepatic Oil Red O staining analysis. Red circle drops are fat droplets.

**Figure 2 animals-13-01820-f002:**
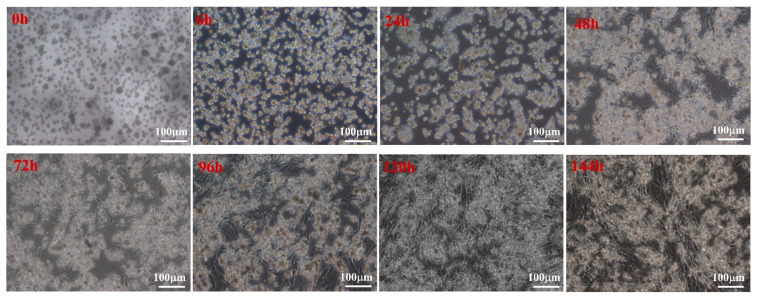
Morphological characteristics of duck embryo hepatocytes at different stages cultured in vitro. At 0 h, primary cells could be observed as bright spots with a round shape. At 6 h, cells adhered to the plates. On day one, hepatocytes began to aggregate and proliferate. At 2 days, hepatocytes tended to spread out and were shaped like an island. At 3 days, hepatocytes ended the proliferation, and the endothelial cells began to proliferate. At 4 days, the activities of endothelial cells increased rapidly. At 5 days, hepatocytes and endothelial cells interlaced similar to a net shape, and activities of the cells began to decrease. At 6 days, activities of the cells decreased even further.

**Figure 3 animals-13-01820-f003:**
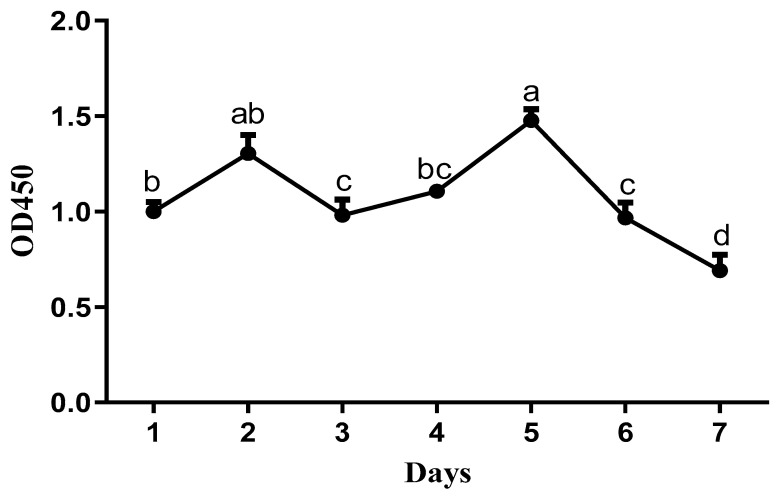
Growth curve of cells (2 × 106 cells/well) cultured for 7 days (*n* = 6). ^a,b,c,d^ The dots with different lowercase letters indicate a significant difference (*p* < 0.05), analyzed by one-way ANOVA followed by Tukey’s multiple comparisons test.

**Figure 4 animals-13-01820-f004:**
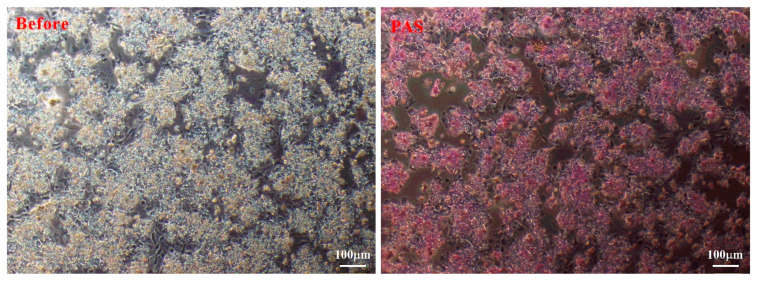
Representative images of duck embryo hepatocytes before (**left**) and after periodic acid-Schiff staining (**right**).

**Figure 5 animals-13-01820-f005:**
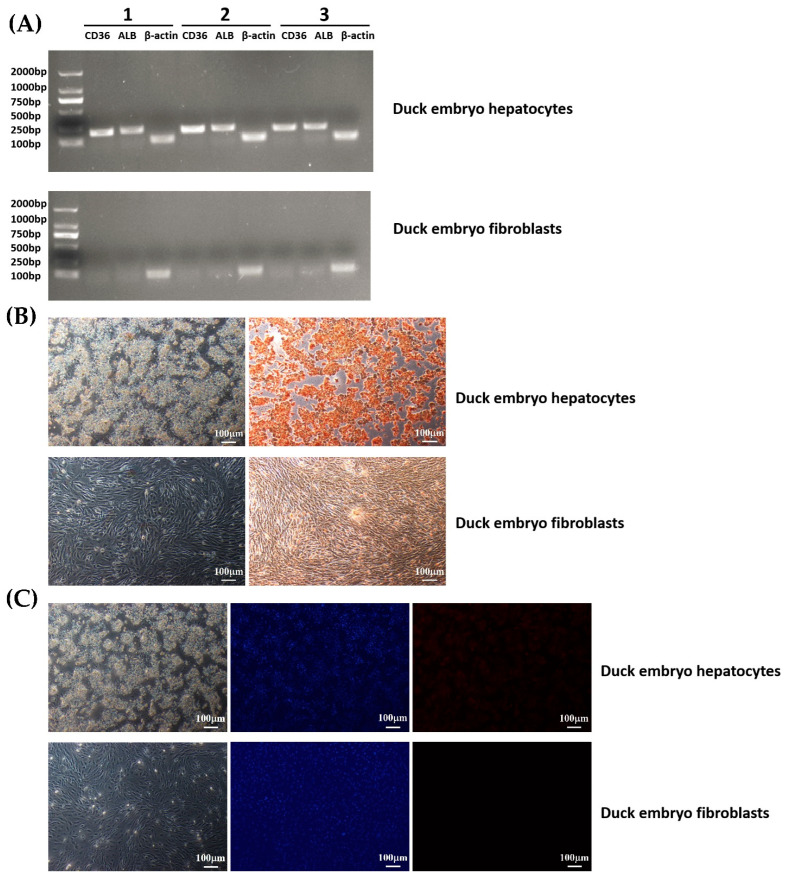
CD36 and ALB expression in duck embryo hepatocytes and fibroblasts. Hepatocytes were isolated from the testes of 21-day-old duck embryos. All assays were carried out 48 h after the isolation and culture of cells. (**A**) Reverse transcription and PCR verification: PCR products were analyzed by electrophoresis on 2.0% (*w/v*) agarose gels. β-actin was used as an internal control. (**B**) IHC assays: The duck embryo hepatocytes cells were all positive for ALB, whereas duck embryo fibroblasts were negative. (**C**) IFA assays: Most duck embryo hepatocytes cells were positive for CK18, while duck embryo fibroblasts were all negative. Nuclei were stained with DAPI.

**Table 1 animals-13-01820-t001:** The primers used in a quantitative RT-PCR assay for gene expression.

Primers	Sequence (5′–3′)	Accession Number	Product Length (bp)
CD36-F	AGCCCAAATGAGAAGGAACA	XM_038183702.1	194
CD36-R	GCATCCCCAATAACAGCAGA		
ALB-F	ACTGCCCTCCATTTTCCTG	NM_001310394.1	202
ALB-R	TCTGTTGTCCTCGTATTCCTTG		
β-actin-F	ATGTCGCCCTGGATTTCG	EF667345.1	165
β-actin-R	CACAGGACTCCATACCCAAGAA		

**Table 2 animals-13-01820-t002:** Length and weight of duck embryo livers.

Age	E14	E21	E27	D7	*p* Value
Length (cm)	1.23 ± 0.07 ^a^	2.06 ± 0.07 ^b^	2.23 ± 0.07 ^c^	4.18 ± 0.11 ^d^	0.000
Weight (g)	0.22 ± 0.03 ^a^	0.64 ± 0.02 ^b^	1.07 ± 0.06 ^c^	5.19 ± 0.23 ^d^	0.000

Values are presented as mean ± SEM. ^a,b,c,d^ Values with different superscript lowercase letters in the same line indicate a significant difference (*p* < 0.05), analyzed by one-way ANOVA followed with Tukey’s multiple comparisons test.

## Data Availability

All data generated or analyzed during this study are included in this published paper.

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
