# Peer review of "Developmental Changes of Duckling Liver and Isolation of Primary Hepatocytes"

_animals, 2023, doi:10.3390/ani13111820_

Round 1

Reviewer 1 Report

Major comments:

Authors said that the present study analyzed the developmental pattern of ducking livers during the embryonic period and the first week after hatching, indicating that the size and weight of the livers increased from E14 to the hatching day. Reviewer cannot see any significance of these data but confirmation.  In general, current standard monolayer hepatocyte cultures are functional in culture for 5 to 7 days. Please discuss details for special methods and novel insights of the duck hepatocyte culture of the present study in Discussion.

Discussion lines 277-282 should be in the introduction.

Please justify CD36 as a hepatocyte marker in Introduction.

Magnification scales are not clear and there are unnecessary characters in the histological figures (Fig2, 4, and 5).

Minor comments:

Line 145: GAPDH should be beta-actin, based on the accession number and product length in Table 1 and Figure 5A.

Table 1: product length (bp) of CD36 maybe 194bp.

Line 201: Dynamic observation? All living tissue from embryo stages are dynamic. Please provide any specific / unique observations of this study.

Line 233: RT-qPCR Figure 5 A is not the qPCR (quantitative PCR)

Line 237: please define CK18 (keratin 18).

Line 241: GAPDH needs to be changed into beta-actin.

Line 266: need references.

Please correct the superscript in the expression of 10 to the power of x in lines 126, 121, 122.

Line 45: more 12than 95%?

Line 69: not clear in “mechanisms of duck hepatocytes”

Line 112: 0.3 um

Line 162: DEHs

Line 179: please correct “Turkey” into “Tukey”.

Line 213: V Morphologic?

Author Response

Point 1: Authors said that the present study analyzed the developmental pattern of ducking livers during the embryonic period and the first week after hatching, indicating that the size and weight of the livers increased from E14 to the hatching day. Reviewer cannot see any significance of these data but confirmation.  In general, current standard monolayer hepatocyte cultures are functional in culture for 5 to 7 days. Please discuss details for special methods and novel insights of the duck hepatocyte culture of the present study in Discussion.

Response 1: Thanks very much for taking your time to review this manuscript. We really appreciate for your constructive comments. We have carefully considered the suggestion of reviewer and made the Introduction and Discussion section much more detailed. In rewriting the Discussion, we have attempted to outline the novel insights of the duck hepatocyte culture to more clearly communicate its importance. In the previous years, researchers attempted to isolate hepatocytes from the liver by mechanical and enzyme digestive methods. However, these methods have great damage to cells, and cells exhibited small number and low viability (Lines 290-293, page 9). Moreover, there are few studies on the isolation of duck embryonic hepatocytes (Lines 328-329, page 10). In our study, the hepatocytes we isolated during the embryonic period are in good condition and function normally. This study will benefit the further development of related work on duck lipid metabolism and virus infection in the future (Lines 335-338, page 10).

These changes will not influence the content and framework of the paper. Meanwhile, all of your questions were answered one by one, and hope that the correction will meet with approval.

Once again, thank you very much for your comments and suggestions.

Point 2: Discussion lines 277-282 should be in the introduction.

Response 2: Thanks for your constructive suggestion, which is highly appreciated. We have carefully scrutinized the manuscript, and made corresponding revisions. These sentences have now been added to the Introduction.

To make this part clearer, we modified this part and now it reads as:

Introduction: “According to previous studies, most hepatocytes expressed Albumin (ALB) and Cytokeratin 18 (CK18). Among them, Albumin (ALB) is highly expressed in the liver and is responsible for a variety of biological functions due to its complex structure. Cytokeratin 18 (CK18) is a cytoskeletal protein and a major intermediate filament family member expressed in the liver. Multiple studies have confirmed that CK18 is released in the liver as one of the most widely studied biomarkers in nonalcoholic steatohepatitis. Moreover, fatty acid translocase, also known as CD36, is abundantly expressed in the liver. Some studies have documented that CD36 is a well-recognized scavenger receptor for fatty acid uptake, which plays multiple physiological roles, including receptor-mediated free fatty acid uptake (Lines 69-80, page 2).”

Point 3: Please justify CD36 as a hepatocyte marker in Introduction.

Response 3: Thank you for your precious comments and advice. Those comments are all valuable and very helpful for revising and improving our paper, as well as the important guiding significance to our researches. For accurate descriptions, we have rephrased these sentences.

The manuscript was modified as follows:

Introduction: “Fatty acid translocase, also known as CD36, is abundantly expressed in the liver. Some studies have documented that CD36 is a well-recognized scavenger receptor for fatty acid uptake, which plays multiple physiological roles, including receptor-mediated free fatty acid uptake. Meanwhile, the presentation of the CD36 has also been modified accordingly (Lines 75-80, page 2).”

Thank you once again for pointing this out, which is valuable for improving the accuracy of the manuscript.

Point 4: Magnification scales are not clear and there are unnecessary characters in the histological figures (Fig2, 4, and 5).

Response 4: We sincerely thank the reviewer for thoroughly examining our manuscript and providing very helpful comments to guide our revision. We have revised it correspondingly in the revised manuscript. The scale bars have been added as requested in figures (Fig 1, 2, 4, and 5). Besides, we have removed the unnecessary characters in the histological figures.

Point 5: GAPDH should be beta-actin, based on the accession number and product length in Table 1 and Figure 5A.

Response 5: We apologize for the mistakes in the manuscript and also carefully checked the entire manuscript. According to the reviewer’s comment, the manuscript has been revised (Line 154, page 4; Line 255, page 8).

Point 6: Table 1: product length (bp) of CD36 maybe 194bp.

Response 6: We are extremely grateful to reviewer for pointing out this problem. You are correct, and we have revised the “195bp” to “194bp” in Table 1 (Line 159, page 4).

Point 7: Line 201: Dynamic observation? All living tissue from embryo stages are dynamic. Please provide any specific / unique observations of this study.

Response 7: Thanks for pointing out the problem. You are correct, all living tissue from embryo stages are dynamic. Thus, “Dynamic observation” may be an inaccurate expression. Here, we found that the duck embryo hepatocytes cultured for seven days exhibit limited proliferative capacity. After two days of culture, typical parenchymal morphology with a characteristic polygonal shape appeared, which was similar to the morphological structure of chicken hepatocytes (Lines 221-222, page 6).

Moreover, the title is now revised to “Morphological observation and growth curve of duck embryo hepatocytes” and modify that part of the Results/Discussion section in the present manuscript to better convey the message of the manuscript (Line 212, page 6; Lines 301-304, page 9).

Finally, based on the suggestion of the reviewer, we have provided more specific observation of this study. We concluded that the hepatocytes were cultured for two days, grown well, exhibited typical parenchymal morphology, and had characteristic polygonal.

Point 8: Line 233: RT-qPCR Figure 5 A is not the qPCR (quantitative PCR)

Response 8: Thank you for your suggestions. We have carefully scrutinized the manuscript, and made corresponding revisions. The new sentence reads as follows.

Reverse transcription and PCR verification. PCR products were analyzed by electrophoresis on 2.0% (w/v) agarose gels. b-actin was used as an internal control (Lines 254-256, page 8).

Point 9: Line 237: please define CK18 (keratin 18).

Response 9: Thanks very much for the professional suggestions. The definition of CK18 (keratin 18) is currently provided in the Introduction (and removed from the discussion) as follows:

“Cytokeratin 18 (CK18) is a cytoskeletal protein and a major intermediate filament family member expressed in the liver. Multiple studies have confirmed that CK18 is released in the liver as one of the most widely studied biomarkers in nonalcoholic steatohepatitis (Lines 73-76, page2)”

Point 10: Line 241: GAPDH needs to be changed into beta-actin.

Response 10: We apologize for the mistakes in the manuscript and also carefully checked the entire manuscript. According to the reviewer’s comment, the manuscript has been revised (Line 145, page 4; Line 241, page 8).

Point 11: Line 266: need references.

Response 11: We deeply appreciate the reviewer’s suggestion. To express this more clearly, we have rephrased the sentences containing this expression and added the reference in the revised manuscript according to the reviewer’s comment.

Hepatocytes exhibited a lower differentiation ability and relatively simpler function during the embryonic stage compared to that during the early postnatal period (Lines 73-76, page2).

Some of the related references are shown as follows.

Okuyama, S.; Kawamura, F.; Kubiura, M.; Tsuji, S.; Osaki, M.; Kugoh, H.; Oshimura, M.; Kazuki, Y.; Tada, M. Real-time fluorometric evaluation of hepatoblast proliferation in vivo and in vitro using the expression of CYP3A7 coding for human fetus-specific P450. Pharmacology research & perspectives 2020, 8, e00642, doi:10.1002/prp2.642.

Yeoh, G.C.; Bennett, F.A.; Oliver, I.T. Hepatocyte differentiation in culture. Appearance of tyrosine aminotransferase. The Biochemical journal 1979, 180, 153-160, doi:10.1042/bj1800153.

Point 12: Please correct the superscript in the expression of 10 to the power of x in lines 126, 121, 122.

Response 12: We sincerely thank the reviewer for thoroughly examining our manuscript and providing very helpful comments to guide our revision. We have revised these correspondingly in the revised manuscript (Lines 129-131 and 135, page 3).

Point 13: Line 45: more 12than 95%?

Response 13: Thank you for your careful review. We are very sorry for the mistakes in this manuscript and inconvenience they caused in your reading. We have modified this sentence as “In poultry, the fat synthesized by the liver accounts for more than 95% of the body fat, which may be related to the key role of the liver (Lines 44-45, page 2).”

Point 14: Line 69: not clear in “mechanisms of duck hepatocytes”

Response 14: Thank you for your precious comments and advice. Those comments are all valuable and very helpful for revising and improving our paper, as well as the important guiding significance to our researches. Moreover, we have rephrased the sentences containing this expression and added the reference in the revised manuscript.

One additional reference was provided and the manuscript was modified as follows: “Although significant progress has been made in understanding the mechanisms underlying the lipid metabolic in geese, while it was not clear in ducks (Lines 67-68, page 2).”

Lu, L.; Chen, Y.; Wang, Z.; Li, X.; Chen, W.; Tao, Z.; Shen, J.; Tian, Y.; Wang, D.; Li, G.; Chen, L.; Chen, F.; Fang, D.; Yu, L.; Sun, Y.; Ma, Y.; Li, J.; Wang, J. The goose genome sequence leads to insights into the evolution of waterfowl and susceptibility to fatty liver. Genome Biol. 2015, 6, 89. doi: 10.1186/s13059-015-0652-y.

Point 15: Line 112: 0.3 um

Response 15: We apologize for the mistakes in the manuscript and also carefully checked the entire manuscript for typographic, grammatical and formatting errors. According to the reviewer’s comment, the manuscript has been revised (Line 121, page 3).

Point 16: Line 162: DEHs

Response 16: We thank the reviewer for this helpful suggestion and modified the text accordingly (Line 171, page 4).

Point 17: Line 179: please correct “Turkey” into “Tukey”.

Response 17: Thank you for your careful review. We are very sorry for the mistakes in this manuscript and inconvenience they caused in your reading. According to your comment, we have changed the "Turkey" to “Tukey”.

And this sentence was modified as follows: “To determine significant differences between the groups, a one-way ANOVA with Tukey test was used, and P < 0.05 was considered statistically significant (Lines 187-189, page 5).”

Point 17: Line 213: V Morphologic?

Response 17: We thank the reviewer for this careful inspection. These writing errors have been corrected (Line 224, page 6).

Finally, we really appreciate all your comments and suggestions. Those comments are all valuable and very helpful for revising and improving our paper, as well as the important guiding significance to our researches. According to your advice, we amended the relevant part in manuscript. All of your questions were answered one by one, and sincerely hope that the correction will meet with approval. Finally, this manuscript has been edited by a professional language editor (Wiley Editing Services) to ensure accuracy and clarity.

Once again, thank you very much for your comments and suggestions.

Reviewer 2 Report

This paper addresses the limited understanding of liver development and primary hepatocyte isolation in ducks, an important species for poultry farming. The authors observed liver developmental changes in ducklings and successfully cultured and maintained primary duck embryonic hepatocytes in vitro. The findings have implications for the study of lipid metabolism in ducks and improve our understanding of avian liver cells.

Although the authors have presented findings on liver development and hepatocyte isolation in ducks, there is a noticeable omission of any comparison with research conducted on similar topics in other avian species, such as ducks. It is therefore my belief that this manuscript falls short of making a significant contribution to the current understanding in this field. Moreover, the lack of reference to previous studies investigating the same topic in other avian species represents a major limitation. In order to enhance the value of the paper, it would be beneficial for the authors to tie their findings to larger implications for poultry farming or human liver diseases, drawing on past research.

Major concerns:

Due to their susceptibility to DHBV infection, primary duck hepatocytes (PDH) are a highly useful cell model for investigating the critical characteristics of hepadnaviral infection. The technique for culturing duck hepatocytes has been thoroughly developed and refined over the years and is now highly mature. However, despite claiming to “provide novel insights for in vitro duck hepatocyte culture”, this study fails to present any original or superior results when compared to previous research involving primary liver cell cultures. Moreover, the authors deliberately avoid discussing and comparing their primary duck hepatocyte culture methodology with those used in previous studies, thereby disregarding the contributions made by their predecessors in the field of duck liver cell culture.

As the primary focus of the paper, the investigation of liver tissue's various developmental stages only includes analysis of four timepoints using H&E and Oil Red O staining, which fails to deliver any novel or useful outcomes. Consequently, the manuscript is deficient in experimental evidence and fails to make significant contributions to the scientific literature.

Additionally, the manuscript lacks a clear research objective or hypothesis, and the discussion section is poorly organized and does not present a clear and coherent set of conclusions. The study's limitations in methodology and analysis, coupled with the lack of comparison with previous literature or studies, also make it difficult for readers to contextualize the findings.

Minor concerns:

1. There is an error on line 45 that reads "for more 12than 95% of the body fat."

2. Overall, the quality of the images in the manuscript is subpar. For example, in Figure 2, there is inconsistency in brightness and contrast between the images, making them appear uncoordinated and unpleasing. Moreover, the DAPI and immunofluorescence images in Figure 5(C) are also unclear.

3. The primary liver cells used in the experiment were a mixture of cell populations. Therefore, the experimental design in section 3.2, which employed the CCK-8 assay to evaluate the viability of primary liver cells, might be questionable.

4. It is unclear on which day the cells in the stained images in Figure 5 were cultivated, and why fibroblast cells were used as a negative control instead of endothelial cells to show albumin and CK18 expression in duck hepatocytes.

Overall, the quality of the English language in the manuscript is average. While it could be improved in some areas, it is adequate for understanding the authors' message. Some minor grammatical and spelling errors were found, but they do not significantly hinder comprehension.

Author Response

Point 1: This paper addresses the limited understanding of liver development and primary hepatocyte isolation in ducks, an important species for poultry farming. The authors observed liver developmental changes in ducklings and successfully cultured and maintained primary duck embryonic hepatocytes in vitro. The findings have implications for the study of lipid metabolism in ducks and improve our understanding of avian liver cells. Although the authors have presented findings on liver development and hepatocyte isolation in ducks, there is a noticeable omission of any comparison with research conducted on similar topics in other avian species, such as ducks. It is therefore my belief that this manuscript falls short of making a significant contribution to the current understanding in this field. Moreover, the lack of reference to previous studies investigating the same topic in other avian species represents a major limitation. In order to enhance the value of the paper, it would be beneficial for the authors to tie their findings to larger implications for poultry farming or human liver diseases, drawing on past research.

Response 1: Thanks very much for taking your time to review this manuscript. We really appreciate for your constructive comments. We have carefully considered the suggestion of reviewer and made the Introduction and Discussion section much more detailed. In rewriting the Discussion, we have attempted to outline the value of the paper to more clearly communicate its importance.

We have revised the Discussion as follows:

In the previous years, researchers attempted to isolate hepatocytes from the liver by mechanical and enzyme digestive methods. However, these methods have great damage to cells, and cells exhibited small number and low viability (Lines 290-293, page 9).

Moreover, there are few studies on the isolation of duck embryonic hepatocytes. According to the previous report of Jia et al., they selected 1-year-old Shaoxing ducks to investigate the replication of DHBV in primary duck hepatocytes. In chickens, primarily tumorigenic cell lines, such as the immortalized chicken embryo liver cell line or embryonic liver cell cultures, have been utilized. Primary duck embryonic hepatocytes are frequently employed for viral propagation. Currently, only immortalized duck embryo liver mesenchymal cells are available from the ATCC, and there are no parenchymal duck liver cell lines. In conclusion, the hepatocytes we isolated during the embryonic period are in good condition and function normally. This study will benefit the further development of related work on duck lipid metabolism and virus infection in the future (Lines 328-338, page 10).

Finally, more references have been added in the revised manuscript. These changes will not influence the content and framework of the paper. Meanwhile, all of your questions were answered one by one, and hope that the correction will meet with approval.

Once again, thank you very much for your comments and suggestions.

Point 2: Due to their susceptibility to DHBV infection, primary duck hepatocytes (PDH) are a highly useful cell model for investigating the critical characteristics of hepadnaviral infection. The technique for culturing duck hepatocytes has been thoroughly developed and refined over the years and is now highly mature. However, despite claiming to “provide novel insights for in vitro duck hepatocyte culture”, this study fails to present any original or superior results when compared to previous research involving primary liver cell cultures. Moreover, the authors deliberately avoid discussing and comparing their primary duck hepatocyte culture methodology with those used in previous studies, thereby disregarding the contributions made by their predecessors in the field of duck liver cell culture. As the primary focus of the paper, the investigation of liver tissue's various developmental stages only includes analysis of four timepoints using H&E and Oil Red O staining, which fails to deliver any novel or useful outcomes. Consequently, the manuscript is deficient in experimental evidence and fails to make significant contributions to the scientific literature. Additionally, the manuscript lacks a clear research objective or hypothesis, and the discussion section is poorly organized and does not present a clear and coherent set of conclusions. The study's limitations in methodology and analysis, coupled with the lack of comparison with previous literature or studies, also make it difficult for readers to contextualize the findings.

Response 2: Thank you for your precious comments and advice. Those comments are all valuable and very helpful for revising and improving our paper, as well as the important guiding significance to our researches. We have revised the paper to address your concerns and hope that it is now clearer.

In the Discussion, we further explored the relevant research on the isolation and culture of hepatocytes as follows:

  • We found that the liver lobules are composed of hepatocytes (parenchymal cells) and non-parenchymal cells. Hepatocytes represent nearly 80% of the total liver volume and carry out several essential liver functions. These were consistent with observations in other mammals and poultry. Approximately 70-80% of the liver’s cells are hepatocytes, which play an important role in organism vital activity (Lines 278-283, page 9).
  • In contrast to mammalians, the avian liver has much less connective tissue than the mammalian liver and lacks a true lobular structure. Hepatocytes exhibited a lower differentiation ability and relatively simpler function during the embryonic stage com-pared to that during the early postnatal period. Therefore, a simple hepatocyte separation method for avians is possible. Based on the yield and purity of hepatocytes, 21-day-old duck embryos were selected for hepatocyte isolation (Lines 296-301, page 9).
  • Currently, there are few studies on the isolation of duck embryonic hepatocytes. According to the previous report of Jia et al., they selected 1-year-old Shaoxing ducks to investigate the replication of DHBV in primary duck hepatocytes. In chickens, primarily tumorigenic cell lines, such as the immortalized chicken embryo liver cell line or embryonic liver cell cultures, have been utilized. Primary duck embryonic hepatocytes are frequently employed for viral propagation. Currently, only immortalized duck embryo liver mesenchymal cells are available from the ATCC, and there are no parenchymal duck liver cell lines. In conclusion, the hepatocytes we isolated during the embryonic period are in good condition and function normally (Lines 328-334, page 10).

In a word, we have added these paragraphs in the Discussion section of the revised manuscript presenting this research objective or hypothesis. To address the practical significance of this research and possible applicability of the presented technique, we have now expanded the Discussion to include a specific reference of the hepatocyte culture in vitro. This study will benefit the further development of related work on duck lipid metabolism and virus infection in the future.

Point 3: There is an error on line 45 that reads "for more 12than 95% of the body fat."

Response 3: Thank you for your careful review. We are very sorry for the mistakes in this manuscript and inconvenience they caused in your reading. We have modified this sentence as “In poultry, the fat synthesized by the liver accounts for more than 95% of the body fat, which may be related to the key role of the liver (Lines 44-45, page 2).”

Point 4: Overall, the quality of the images in the manuscript is subpar. For example, in Figure 2, there is inconsistency in brightness and contrast between the images, making them appear uncoordinated and unpleasing. Moreover, the DAPI and immunofluorescence images in Figure 5(C) are also unclear.

Response 4: We sincerely thank the reviewer for thoroughly examining our manuscript and providing very helpful comments to guide our revision. We are sorry for the subpar quality of the images in the manuscript. In the revised manuscript, we have provided clearer images. Meanwhile, the scale bars have been added in figures (Fig 1, 2, 4, and 5). Besides, the inconsistency in brightness and contrast between the images might be due to the different observation times. In future studies, we will try our best to collect images under a constant condition, including the same illumination, background and contrast. Once again, we thank the reviewer for pointing out this issue.

Point 5: The primary liver cells used in the experiment were a mixture of cell populations. Therefore, the experimental design in section 3.2, which employed the CCK-8 assay to evaluate the viability of primary liver cells, might be questionable.

Response 5: Thanks again to the reviewer for pointing out this issue. Here, we would like to further evaluate the cells viability. We are sorry for the incorrect expression and have corrected it. And then, we rephrased these sentences in the Results section and in the Discussion sections.

The results were modified as follows. “As shown in Figure 3, the growth curves of the cells showed that the proliferation activity of the cells was the strongest at the second day and the fifth day. After the fifth day, the viability of the cells decreased significantly (P<0.001).” (Lines 217-220, page 6)

Afterwards, the revised Discussion of the results in combination with morphological observations reads: “Furthermore, we found that cell viability reached its maximum with incubation at the second and fifth day, respectively. We speculate that duck embryo hepatocytes underwent a proliferation period of two days. After that, endothelial cells entered a logarithmic phase for five days, after which the cell's activities began to decrease. In combination, morpho-logical changes of duck hepatocytes were closely related to cell viability.” (Lines 306-310, page 9)

Thanks again to the reviewer for your professional suggestions.

Point 6: It is unclear on which day the cells in the stained images in Figure 5 were cultivated, and why fibroblast cells were used as a negative control instead of endothelial cells to show albumin and CK18 expression in duck hepatocytes.

Response 6: We thank the reviewer for this question. We are sorry for this missing information in Figure 5 and inconvenience they caused in your reading. We have added the following statement: “Hepatocytes were isolated from the testes of 21-day-old duck embryos. All assays were carried out 48 h after the isolation and culture of cells (Lines 252-254, page 5).”

In addition, we carefully considered the selection of negative control before the experiment. In this study, fibroblast cells were used as a negative control instead of endothelial cells to show albumin and CK18 expression in duck hepatocytes.

We now more clearly describe the critical advantages of the fibroblast cell in the discussion, as follows: “In this study, fibroblast cells were used as a negative control. Fibroblast cells, as a common cell type, are easier to isolate and culture. At the same time, studies have demonstrated that fibroblast cells do not express CK18, which allows the specific detection of hepatocytes by the fluorescent signal of CK18.” For these reasons, fibroblast cells are used as a negative control to confirm that the fluorescent signal is specific rather than non-specific binding or other factors.

Of course, we also very much agree with the reviewer's point that the negative control selection with liver endothelial cells might more effectively evaluate the characteristics and functions of hepatocytes. Therefore, in order to make the experimental data more rigorous, the selection of control cells will be more comprehensive considered in future experiments.

We sincerely thank the reviewer for raising this professional question and helpful suggestion.

Point 7: Overall, the quality of the English language in the manuscript is average. While it could be improved in some areas, it is adequate for understanding the authors' message. Some minor grammatical and spelling errors were found, but they do not significantly hinder comprehension.

Response 7: Thank you again for the opportunity to revise our manuscript. Those comments are all valuable and very helpful for revising and improving our paper. We have tried our best to revise our manuscript according to the comments and given responses point by point. We sincerely hope the revised manuscript is now suitable for publication in Animals. Finally, this manuscript has been edited by a professional language editor (Wiley Editing Services) to ensure accuracy and clarity.

Once again, thanks very much for taking your time to review this manuscript.

Reviewer 3 Report

Dear Authors, 

This paper is an excellent attempt to enhance poultry research. 

These are my points:

1. Please add more introduction and background which explains the significance of your study. 

2. Figures are not appropriately labeled. No scale bars in Figure 1.

It would be good if you could add the stain name in the figure for the pictures. 

3. Statistical differences have not been explained adequately. 

4. What are your proposed future studies related to this research, please explain. 

Author Response

Point 1: Please add more introduction and background which explains the significance of your study.

Response 1: Thanks very much for taking your time to review this manuscript. We really appreciate for your constructive comments. We have carefully considered the suggestion of reviewer and made the Introduction and Discussion section much more detailed. In rewriting the Discussion, we have attempted to outline the novel insights of the duck hepatocyte culture to more clearly communicate its importance.

These changes will not influence the content and framework of the paper. Meanwhile, all of your questions were answered one by one, and hope that the correction will meet with approval.

Once again, thank you very much for your comments and suggestions.

Point 2: Figures are not appropriately labeled. No scale bars in Figure 1. It would be good if you could add the stain name in the figure for the pictures.

Response 2: We sincerely thank the reviewer for thoroughly examining our manuscript and providing very helpful comments to guide our revision. We have revised it correspondingly in the revised manuscript. The scale bars have been added as requested in figures (Fig 1, 2, 4, and 5). Besides, we have removed the unnecessary characters in the histological figures. The stain names have also been added as requested.

Point 3: Statistical differences have not been explained adequately.

Response 3: We are extremely grateful to reviewer for pointing out this problem. Those comments are all valuable and very helpful for revising and improving our paper, as well as the important guiding significance to our researches. And we have made some modifications in Statistical analysis.

Modify the following:

 “Values are presented as mean ± SEM. a,b Values with different superscript lowercase letters in the same line indicate a significant difference (P<0.05), analyzed by one-way ANOVA followed with Tukey’s multiple comparisons test (Lines 204-206, page 5).”

Point 4: What are your proposed future studies related to this research, please explain.

Response 4: Thanks very much for the professional questions and helpful suggestions. We address this in the extensively rewritten discussion. According to the reviewer´s valuable suggestions, we have discussed the possible explanations and future study directions in the revised manuscript.

The new additions are as follows:

In the previous years, researchers attempted to isolate hepatocytes from the liver by mechanical and enzyme digestive methods. However, these methods have great damage to cells, and cells exhibited small number and low viability (Lines 290-293, page 9). Moreover, there are few studies on the isolation of duck embryonic hepatocytes (Lines 328-329, page 10). In our study, the hepatocytes we isolated during the embryonic period are in good condition and function normally. This study will benefit the further development of related work on duck lipid metabolism and virus infection in the future (Lines 335-338, page 10).

Finally, we really appreciate all your comments and suggestions. Those comments are all valuable and very helpful for revising and improving our paper, as well as the important guiding significance to our researches. According to your advice, we amended the relevant part in manuscript. All of your questions were answered one by one, and sincerely hope that the correction will meet with approval. Finally, this manuscript has been edited by a professional language editor (Wiley Editing Services) to ensure accuracy and clarity.

Once again, thank you very much for your comments and suggestions.

Round 2

Reviewer 2 Report

The authors have made an appropriate effort to respond to the comments. The revised Discussion section provides a more detailed explanation of the importance of the paper and draws on previous research to support their findings. They have addressed the concern regarding the lack of comparison with similar research in other avian species by referencing previous studies investigating the same topic in other avian species. Overall, these revisions have improved the quality of the manuscript and its value to the field.

However, the quality of Figure 2 is inadequate and fails to meet the standards required for publication. While the authors have acknowledged the need to be more diligent in capturing and presenting clear images in future work, the current image quality is unacceptable. Therefore, I strongly recommend that Figure 2 be replaced with higher quality photographs. Once this issue has been resolved, I will have no further concerns with the manuscript.

Author Response

Point 1: The authors have made an appropriate effort to respond to the comments. The revised Discussion section provides a more detailed explanation of the importance of the paper and draws on previous research to support their findings. They have addressed the concern regarding the lack of comparison with similar research in other avian species by referencing previous studies investigating the same topic in other avian species. Overall, these revisions have improved the quality of the manuscript and its value to the field.

However, the quality of Figure 2 is inadequate and fails to meet the standards required for publication. While the authors have acknowledged the need to be more diligent in capturing and presenting clear images in future work, the current image quality is unacceptable. Therefore, I strongly recommend that Figure 2 be replaced with higher quality photographs. Once this issue has been resolved, I will have no further concerns with the manuscript.

Response 1: We thank the reviewer for reviewing our revised manuscript again. Figure 2 have been replaced with higher quality photographs, and hope that the correction will meet with approval.

Once again, thank you very much for your comments and suggestions.
